



# Reconciling conflicting evidence for the cause of the observed early 21st century Eurasian cooling

Stephen Outten[1,2,*], Camille Li[3,2,*], Martin P. King[3,4,2], Lingling Suo[1,2], Peter Y. F. Siew[3,2], Richard Davy[1,2], Etienne Dunn-Sigouin[3,2], Shengping He[3,2], Hoffmann Cheung[5,6,2], Erica Madonna[3,2], Tore Furevik[1,2], Stefan Sobolowski[4,2], Thomas Spengler[3,2], and Tim Woollings[7]

[1]Nansen Environmental and Remote Sensing Center, Bergen, Norway
[2]Bjerknes Centre for Climate Research, Bergen, Norway
[3]Geophysical Institute, University of Bergen, Bergen, Norway
[4]NORCE Norwegian Research Centre, Bergen, NORWAY
[5]School of Atmospheric Sciences & Guangdong Province Key Laboratory for Climate Change and Natural Disaster Studies, Sun Yat-sen University, Zhuhai, China
[6]Southern Marine Science and Engineering Guangdong Laboratory (Zhuhai), Zhuhai, China
[7]Atmospheric, Oceanic and Planetary Physics, University of Oxford, Oxford, United Kingdom
[*]These authors contributed equally to this work.

**Correspondence:** Stephen Outten (stephen.outten@nersc.no)

**Abstract.** It is now well established that the Arctic is warming at a faster rate than the global average. This warming, which has been accompanied by a dramatic decline in sea ice, has been linked to cooling over the Eurasian subcontinent over recent decades, most dramatically during the period 1998-2012. This is a counterintuitive impact under global warming given that land regions should warm more than ocean (and the global average). Some studies have proposed a causal teleconnection from

Arctic sea ice retreat to Eurasian wintertime cooling; other studies argue that Eurasian cooling is mainly driven by internal variability and the relationship to sea ice is coincidental. Overall, there is an impression of strong disagreement between those holding the "ice-driven" versus "internal variability" viewpoints. Here, we offer an alternative framing showing that the sea ice and internal variability views can be compatible. Key to this is viewing Eurasian cooling through the lens of dynamics (linked primarily to internal variability with a small contribution from sea ice; cools Eurasia) and thermodynamics (linked to sea ice

retreat; warms Eurasia). This approach, combined with recognition that there is uncertainty in the hypothesized mechanisms themselves, allow both viewpoints (and others) to co-exist and contribute to our understanding of Eurasian cooling. A simple autoregressive model shows that Eurasian cooling of this magnitude is consistent with internal variability, with some periods being more susceptible to strong cooling than others. Rather than posit a "yes-or-no" causal relationship between sea ice and Eurasian cooling, a more constructive way forward is to consider whether the cooling trend was more likely given the observed

sea ice loss, as well as other sources of low-frequency variability. Taken in this way both sea ice and internal variability are factors that affect the likelihood of strong regional cooling in the presence of ongoing global warming.



# 1 Introduction

Global mean temperature has increased by over 1°C since pre-industrial times due to anthropogenic forcing (Masson-Delmotte et al., 2021). Nowhere has this change been more pronounced than in the Arctic, where temperatures have risen two to four
times as fast as the global average, with some studies suggesting even faster rates (Huang et al., 2017; Eldevik et al., 2020). Against this backdrop, it is somewhat surprising to note a wintertime cooling trend in Eurasia over the last decades (Figure 1 and Outten and Esau, 2012) coinciding with a number of cold, harsh winter seasons (Vihma, 2014; Cohen et al., 2014; Kug et al., 2015; Overland et al., 2016). Such continental regions are actually expected to exhibit stronger warming than the global average owing to differences in how the vertical temperature profile in the atmosphere adjusts to radiative forcing over land
versus ocean (Joshi et al., 2008; Sherwood et al., 2020, and references therein). Although counter-intuitive, some studies have suggested that Eurasian cooling may be part of the climate system's response to global warming, a regional manifestation of a hemispheric Warm Arctic–Cold Continents pattern (e.g., Overland et al., 2011) and a direct consequence of sea ice retreat in the Barents-Kara Sea (e.g., Honda et al., 2009; Petoukhov and Semenov, 2010; Semenov and Latif, 2015). Others have voiced reservations over whether such a link tying midlatitude weather and climate to Arctic warming exists (e.g., McCusker et al.,
2016; Blackport et al., 2019; Cohen et al., 2020).

Various mechanisms have been proposed to explain how Arctic warming might cause Eurasian cooling. Most hinge on the fact that sea ice retreat exposes the atmosphere to the relatively warm ocean, which acts as a heat source, causing changes to the large-scale atmospheric circulation that have remote midlatitude impacts. Suggested pathways include the following, the first three of which act to enhance the wintertime anticyclone that brings cold Arctic air into continental Eurasia:

1. The oceanic heat source triggers a Rossby wave train that enhances the anticyclone over Eurasia (Honda et al., 2009; Kug et al., 2015).

   2. The oceanic heat source weakens the stratospheric polar vortex, which leads to a strengthening of the anticyclone over Eurasia (Cohen et al., 2014; Kim et al., 2014; King et al., 2016; Zhang et al., 2018a).

   3. The receding sea ice edge changes local baroclinicity, which affects the characteristics of cyclones/anticyclones, leading
to a strengthening of the anticyclone over Eurasia (Inoue et al., 2012; Zhang et al., 2012).

   4. The weakened equator-to-pole temperature gradient slows down the jet stream, resulting in more atmospheric blocking and more persistent weather patterns in the mid-latitudes (Francis and Vavrus, 2012; Petoukhov and Semenov, 2010; Francis and Vavrus, 2015).

There has also been a suggestion that Arctic warming leads to a split jet stream that can trap atmospheric waves, allowing
weather patterns to stagnate, but this is primarily a summertime phenomenon (Petoukhov et al., 2013; Coumou et al., 2014).

Conversely, a number of recent studies have offered an alternative view: the relationship between Arctic sea ice retreat and Eurasian cooling is not causal. These studies point out that the forced midlatitude response to sea ice perturbations is generally very weak, and argue that the observed correlation between sea-ice and Eurasian winter temperatures arises primarily from





internal climate variability (McCusker et al., 2016; Ogawa et al., 2018; Blackport et al., 2019; Fyfe, 2019; Blackport and
Screen, 2021; Zappa et al., 2021). Furthermore, surface temperature variability is expected to decrease under global warming
(Schneider et al., 2015; Blackport and Kushner, 2016; Holmes et al., 2016), meaning that extremely cold Eurasian winters
should become less frequent in the future.

A definitive answer to if and how sea ice loss has caused Eurasian cooling has proven elusive, in large part because our
mechanistic understanding of Arctic-midlatitude teleconnections remains incomplete (Barnes and Screen, 2015; Wallace et al.,
2016; Shepherd, 2021). Most of our knowledge on how surface forcing alters large-scale atmospheric circulation comes from
midlatitude theories, and we are just starting to investigate whether these apply at higher latitudes as well (Perlwitz et al., 2015;
Sellevold et al., 2016; Hell et al., 2020). The stratosphere-troposphere coupling that is thought to be a key step in many of
the proposed teleconnection pathways is not well understood (Kidston et al., 2015). Furthermore, the observational record is
relatively short, with only 40 years of reliable reanalysis data for the Arctic, where observational data coverage is sparse and
the changes are fastest. Thus we have limited data not only for the period of rapid change, but also for the period preceding it.
This makes it difficult to assess the robustness of teleconnection pathways that may be non-stationary, intermittent, and subject
to large internal variability (Overland et al., 2016). Finally, the question of how much confidence we have in our models further
clouds the issue. Together, these factors have led to disparate and often conflicting opinions on the role of sea ice for the
observed Eurasian cooling.

How can one make sense of the large body of literature on Eurasian cooling? A recent, comprehensive review about Arctic
warming and mid-latitude weather (not just focused on Eurasia) categorized existing studies into two main groups, observa-
tional studies in support of a causal linkage, and modelling studies arguing for little or no connection (Cohen et al., 2020).
This broad grouping is a useful starting point, but this review takes an alternative approach based on the fact that the separation
between observational and modelling studies is not as clear-cut for the regional case of Eurasian cooling. There are modelling
studies that support the existence of a linkage, and observational studies that do not. Even among studies that arrive at appar-
ently different conclusions, the results themselves often exhibit substantial overlap that we feel has not been given adequate
attention. By highlighting this common ground, and seeing where and why various studies diverge in their interpretations, we
arrive at a summarizing framework that brings together insights from a side range of work.

The rest of the paper is organized as follows. Section 2 reviews the signatures of Eurasian cooling, Barents-Kara sea ice loss,
and atmospheric circulation variability over the satellite period to establish the observational basis for the linkage. Section 3
summarizes modelling studies investigating the teleconnection, including the relevant spatial patterns, timing and mechanisms.
Section 4 explores ways to reconcile studies that come to seemingly opposite conclusions while section 5 proposes a new
framework for understanding the divergent views. Overall, this paper aims to summarize what we have learned about the role
of sea ice in Eurasian cooling to date, examine areas of agreement and disagreement in the scientific literature more carefully,
answer why the debate remains unresolved and suggest fruitful directions for future research.





## 2 Eurasian Cooling in Observations and Reanalyses

Much of the debate regarding Eurasian Cooling revolves around the possible linkages between the observed decline of wintertime near-surface air temperature over Central Eurasia and the observed decline in Arctic sea ice extent. Proposed linkages primarily involve changes in the large-scale atmospheric flow to act as a teleconnection between the two. Thus, we start by presenting observed trends in near-surface temperature, sea ice extent, and the large-scale flow before investigating the proposed mechanisms for their possible interconnection, we present the observed trends in these three fields.

### 2.1 Temperature

Eurasian cooling and Arctic warming are apparent in the trends of wintertime temperature over the past two decades. Reanalyses assimilate temperature observations and provide a high-quality gridded dataset, making them the best data source available for investigating temperature trends. Reanalyses show warming over the Arctic that extends into the upper troposphere, especially over regions of sea ice reduction, although the upper tropospheric signature is not well constrained given the lack of observations at altitude in this remote location (Screen et al., 2012; Screen and Simmonds, 2010b). Here we focus on the trends in 2-metre temperature for the winter months of December-January-Febraury in the ERA5 reanalysis (Hersbach et al., 2020) provided by the European Centre for Medium Range Forecasting (ECMWF). The trends over the period 1998 to 2012, the period with the strongest cooling trends, are shown for the Northern Hemisphere (Figure 1). Strong positive trends in excess of 4 K/decade are seen over the Barents-Kara-Laptev Seas and Eastern Siberian Sea, extending into the Arctic Ocean, accompanied by negative temperature trends of over -4 K/decade over a large area of Central Eurasia (southern Russia, Kazakhstan, northern Uzbekistan, extending southeast over Mongolia and Northern China). This large area of cooling contributed to the global warming hiatus during the first decade of this century (Kaufmann et al., 2011; Cohen et al., 2012; Li et al., 2015), but unlike the North American part of the hiatus signal, it has not been linked to tropical Pacific forcing (Kosaka and Xie, 2013). This pattern of Warm Arctic-Cold Continent (WACC), or more specifically, Warm Arctic-Cold Eurasia (WACE) has been identified in other observational and reanalysis datasets, including the previous generation of ECMWF reanalysis, ERA-Interim (Mori et al., 2014; Outten and Esau, 2012; Chen et al., 2018), the National Center for Environmental Prediction–National Center for Atmospheric Research (NCEP-NCAR) reanalysis (Overland et al., 2011; Inoue et al., 2012; Outten et al., 2013), the Goddard Institute for Space Studies Surface Temperature Analysis, GISTEMP (McCusker et al., 2016), and the Hadley Centre-Climatic Research Unit global temperature dataset, HadCRUT4 (Shu et al., 2018).

Not all studies have identified such strong or significant cooling trends over the Eruasian sector (e.g., Ogawa et al., 2018). The main issue is that the trend depends on the time period over which it is calculated (and to a lesser extent on the region, Supplemental Figure S2). This means that two studies using the same methods and data to identify Eurasian cooling may still disagree over the presence, location and strength of Eurasian cooling based solely on the choice of the study period. To illustrate the transient nature of Eurasian cooling, Figure 2 shows the trends in wintertime 2-metre temperature over the Eurasian region for a range of starting years and period lengths. The greatest cooling trends are found for a 15-year long period starting in 1998, that is, covering the period of 1998 to 2012 (as in Figure 1). Maintaining the period length of 15 years but shifting it just three





years earlier (1995) or later (2001) removes almost all of the significant cooling in Eurasia. Similarly, maintaining the starting
year as 1998 but extending the period to 20 years again removes the significant Eurasian cooling. The sensitivity of Eurasian
cooling to increasing period length or later start year suggests that it is a transient phenomenon which has already passed, and
hence more likely related to decadal variability than global warming (Blackport and Screen, 2020).

Figure 2 shows two other features of note. Firstly, the diagonal band of panels with pronounced Eurasian cooling sits
within a broader band exhibiting significant Arctic warming. Any period with significant Eurasian cooling also has significant
Arctic warming, giving the classic WACE pattern. But Arctic warming is also apparent just before and after the occurrence of
Eurasian cooling. For example, for 15-year trends, strong Arctic warming starts to appear in 1991 and increases in area and
magnitude towards the mid-90s, when significant cooling begins to appear over Central Eurasia. This pattern of warming and
cooling peaks in 1998, after which both the Eurasian cooling and the Arctic warming decrease in magnitude and area until the
Eurasian cooling disappears completely in 2001, and the Arctic warming trends disappear in the period starting in 2005. This
is suggestive of a link between the Arctic warming and Eurasian cooling, however, this analysis cannot tell us whether the link
is causal or the result of internal variability in the climate system.

Secondly, Figure 2 shows other smaller regions of weak cooling not associated with Eurasian cooling. Even in a warming
world, there could be periods of cooling over small regions somewhere on the planet. Such cooling is visible in Figure 2 and
elsewhere in the Northern Hemisphere (not shown), for example, over Scandinavia for periods of length 15 to 17 years, starting
in 1988 to 1989, or in the Russian Arctic for periods of length 15 to 19 years, starting in 1983/1984 to 1980/1981, respectively.
These isolated cooling signals are smaller in area, weaker in magnitude, and shorter lived than the Eurasian cooling, with no
associated region of concurrent intense warming, all of which suggest that they are most likely generated by internal variability.

When examining trends over periods as short as fifteen years, outlying individual events can heavily bias the trend calcula-
tion. In the case of Eurasian cooling, it has been suggested that a few individual extreme cold winters may have given rise to
an apparent cooling trend over the period 1998-2012 (Cohen et al., 2020). Trends are often calculated through simple linear
regression using the Ordinary Least Squares method, a parametric method that is very sensitive to outliers. Non-parametric
methods (e.g. Theil-Sen) provide trend estimates which are more robust to the presence of outliers. A comparison of these
two methods for calculating the trends in ERA5 wintertime 2-metre temperatures shows little difference between the resulting
trends, indicating that the strong cooling signal is not caused by the presence of individual extreme winters (Supplemental
Figure S1).

## 2.2    Sea-ice

Arctic sea ice has declined rapidly during recent decades (Stroeve et al., 2012). From 1979 to 2000, Arctic sea-ice extent (de-
fined as the area covered by sea ice with a concentration of at least 15%) exhibited a seasonal cycle ranging from approximately
4.7 million km$^2$ in September to 13.2 million km$^2$ in March, according to the National Snow and Ice Data Centre [NSIDC]
observations. Since 2000, there have been multiple summers when sea ice area was less than 4 million km$^2$, with the lowest
being a mere 2.4 million km$^2$ in 2012. The trend in Arctic sea-ice extent is sensitive to the season and the selected period
(Onarheim et al., 2018). The long-term (1979-2019) trend in September (annual minimum) sea-ice extent is -0.5 million km$^2$





per decade, peaking at -1.2 million km$^2$ per decade during 1998-2012. Meanwhile the long-term trend in March (annual maximum) is barely -0.04 million km$^2$ per decade, peaking at -0.2 million km$^2$ per decade during 1998-2012 (Supplemental Figure

S3). Sea ice retreat during the peak period was likely enhanced by increased poleward ocean heat transport associated with the positive phase of the Atlantic Multidecadal Variability (Årthun and Eldevik, 2016; Luo et al., 2017). As a result, much of the Arctic ice cover has become first-year ice, which is thinner and less stable than the multi-year ice it replaced (Maslanik et al., 2007). This is reflected in the discrepancy in the long-term trends of March and September sea ice extent, since the winter sea ice largely recovers each year, but is replaced by thinner sea ice which melts more easily, thus leading to a stronger decline in

summer sea ice.

Given the differences in the rate of total sea ice retreat between the summer and winter, there are also large differences in the regional distribution of sea ice loss (Close et al., 2015). While summertime retreat affects most of the Arctic Ocean and surrounding seas (not shown), the wintertime retreat represents a far smaller percentage of the total sea ice area, and is thus restricted to the marginal ice zones, in particular the Greenland-Barents-Kara Seas (Figure 3). Marginal ice zones are

a critical region for atmosphere-ocean interactions. Under normal conditions, the sea surface temperature and near-surface temperature above the sea are very similar, as the atmosphere adjusts to the ocean surface temperature. However, sea ice causes a decoupling of the atmosphere and ocean, allowing the air temperature to be considerably colder in winter. In the marginal ice zone, this decoupling breaks down as cold Arctic air is exposed to the relatively warm ocean surface. Since surface heat fluxes are controlled by the temperature contrast between air and water, the marginal ice zones often see some of the strongest

air-sea fluxes observed (Suo et al., 2016; Papritz and Spengler, 2017). The predominant direction of heat transfer between the atmosphere and ocean can provide clues as to whether the sea ice-Eurasian cooling relationship is causal, and will be discussed further in Section 4.

While the mechanisms by which sea-ice loss may cause Eurasian cooling continue to be debated, one remarkably robust result is that Barents-Kara sea-ice extent and central Eurasian temperatures co-vary over the period of interest. This co-variability

has been established through simple correlation of area-averaged indices from the Barents-Kara Sea and Central Eurasia (Outten and Esau, 2012), Empirical Orthogonal Function (EOF) analysis (Mori et al., 2014), and Singular Value Decomposition (SVD) analysis (Outten and Esau, 2012; Outten et al., 2013). This pattern of co-variability between Arctic sea ice and 2-metre temperature is shown in Figure 4, as the first mode of covariability derived using SVD analysis for the ERA-Interim reanalysis (1989-2010). The sea ice and 2-metre temperature are strongly related to one another in this first mode, as indicated by their

coupling correlation coefficient of r=0.79, and the amount of covariance explained by this mode is approximately 51%. While these studies robustly show a teleconnection in the sense of "a significant positive or negative correlation in the fluctuations of a field at widely separated points" (American Meteorological Society, 2021), many are purely statistical in nature and do not demonstrate a teleconnection in the mechanistic sense of "a linkage between weather changes occurring in widely separated regions of the globe" (American Meteorological Society, 2021).





## 2.3   Large-Scale Flow


The observed Eurasian cooling trend is associated with an anticyclonic circulation anomaly, referred to as Ural blocking, located over the Urals region of northern Eurasia (Figure 1, contours). Ural blocking reinforces the Siberian High, a semi-permanent circulation feature in wintertime that advects cold Arctic air over the continent. The frequency of Ural blocking increased during 1998-2012 (Tyrlis et al., 2020; Chen et al., 2018; Luo et al., 2019), possibly linked to the decreased meridional temperature gradient resulting from increased Arctic warming in this period (Luo et al., 2017, 2018; Yao et al., 2017). This increased blocking is thought to have promoted severe cold winters across central and eastern Eurasia, both by enhancing cold advection and radiative cooling of the continental interior, and by weakening westerly flow and the import of warm, moist air masses (Lu and Chang, 2009; Lu et al., 2010; Zhang et al., 2012; Wang and Chen, 2014). A decrease in cyclone track density over Eurasia has been reported during this period as well, although results depend somewhat upon the methods and data used (Zhang et al., 2012; Neu et al., 2013). The anticyclonic anomaly is not only a surface feature, but also extends through the troposphere, and its effect can be seen in the upper level geopotential height field which shows a positive (negative) anomaly at high (mid) latitudes (Overland and Wang, 2010; Zhang et al., 2012). As with the cooling trends, this circulation anomaly is non-stationary (Petoukhov and Semenov, 2010; Semenov and Latif, 2015), and is sensitive to conditions upstream over the North Atlantic ocean (Luo et al., 2016).



Whether Eurasian cooling is connected to the North Atlantic Oscillation (NAO) is less clear. The North Atlantic Oscillation (NAO) is one of the dominant modes of variability in the Northern Hemisphere (Hurrell, 1995; Pinto and Raible, 2012), and is known to vary on decadal timescales (Woollings et al., 2015). The NAO's role in shaping weather and climate in the Northern Hemisphere is well established (Hurrell et al., 2003), and it has been identified as a possible precursor to Ural blocking (Murto et al., 2022). However, its sensitivity to Arctic sea ice extent is more tenuous (Kolstad and Screen, 2019; Siew et al., 2020, 2021). In fact, the period with the strongest cooling trend over central Eurasia (Figure 2) is associated with a period when the wintertime NAO was largely neutral and exhibited no strong trend (Figure 1 and Blackport and Screen, 2020), suggesting that the NAO is of limited value in explaining the observed Eurasian cooling.



Other changes in large-scale flow have been noted during the period of Eurasian cooling, many of which are consistent with the proposed mechanisms giving rise to the observed Eurasian cooling. These include a weakening of the stratospheric polar vortex, an increase in vertical planetary-scale wave propagation, and changes to Rossby wave trains (Jaiser et al., 2012, 2013; Kim et al., 2014; Nakamura et al., 2015, 2016; Sun et al., 2015; Kretschmer et al., 2016; Hoshi et al., 2017, 2019; Zhang et al., 2018a; Ye and Messori, 2020). In general, the trends in these large-scale features are weak compared with year-to-year variability. Furthermore, the linkage to autumn or winter sea-ice variability is even more difficult to assess given the high degree of cross-correlation between all the signals and the short length of the observational record (Simon et al., 2020). Notably, the circulation trends related to Eurasian cooling have not continued into the most recent decade despite ongoing sea ice loss, thus weakening support for a strong causal linkage (Blackport and Screen, 2020).







## 2.4 Observational Summary

Wintertime near-surface cooling over central Eurasia is well established in the observations, and it is concurrent with strong warming and sea-ice decline over the Barents-Kara Seas from the late 1990s to the early 2010s. Along with the sea-ice loss and Eurasian cooling trends, there are weaker but detectable trends in atmospheric circulation indicating changes in the polar vortex, jet stream, and Rossby waves. That Eurasian cooling occurs when Arctic warming is strong supports the idea of an Arctic-midlatitude link, but not necessarily a causal one. In fact, analyses of the surface heat fluxes that communicate sea ice changes to the atmosphere show that the fluxes - and hence any potential linkages - are strongly modulated by internal variability (Sorokina et al., 2016; Blackport et al., 2019). Furthermore, the circulation trends and Eurasian cooling itself have not continued into the most recent decade, while sea ice loss and Arctic warming unequivocally have. In summary, the clear relationship between sea ice and Eurasian cooling from the late 1990s to the early 2010s has not continued through to the present, leading more recent studies to emphasize the role of internal variability in explaining the observational record.

## 3 Eurasian Cooling in Modelling Studies

The findings, interpretations, and conclusions regarding Eurasian cooling across modelling studies are far more varied than those across observational studies, and it is here that the debate intensifies. Generally speaking, modelling studies can be grouped into three types as follows:

(a) Idealised or simplified model experiments.

(b) State-of-the-art perturbation experiments (uncoupled or coupled) with specially configured forcings and settings.

(c) General-purpose multi-model coupled model ensembles, such as those from the Coupled Model Intercomparison Projects (CMIP).

A general overview of these modelling studies and their main messages appears in section 3.1, followed in section 3.2 by a more in-depth discussion of selected studies that highlight concepts we will use to help reconcile disparate findings (Section 4). We end with a discussion of some limitations of climate models that have direct relevance for the problem of Eurasian cooling in section 3.3. Note that some studies discussed here focus on the atmospheric responses to sea ice loss or variability, but do not in themselves extend to the potential remote impact of Eurasian cooling.

## 3.1 Overview of modelling results by experiment type

Given the large number of modelling studies investigating Arctic-midlatitude teleconnections, it is useful to first summarize the general lessons learned from the three types of modelling studies. Type **a** studies are normally used to probe the mechanisms behind atmospheric responses to high-latitude surface forcings. They show clear but weak midlatitude responses and provide insight into the underlying dynamical processes, but their simplified nature makes it difficult to assess the importance



of the identified linkages in the real climate system (Newson, 1973; Sellevold et al., 2016; Zhang et al., 2018b). Type **b** studies typically involve inducing sea ice loss according to transient scenarios (e.g. historical evolution of sea ice concentration and thickness) or extreme sea ice conditions, in AGCMs or GCMs. These sea ice perturbations can be combined with other adjustments, such as nudging the stratosphere towards its climatology rather than leaving it "free". Many type **b** studies, but

not all, report detectable atmospheric responses including Eurasian cooling, although the signals are usually only statistically significant if one accounts for uncertainties due to internal variability by using a large enough sample, i.e., a large number of simulations (Honda et al., 2009; Liu et al., 2012; Ogawa et al., 2018). Studies using free-running (unperturbed) simulations from coupled climate models (type **c**) report not only a failure to identify robust Eurasian cooling trends, but also an inconsistency in the overall atmospheric signals across models. Many of these models are at the cutting edge of climate science,

and confidence in their fidelity is high, but simulations are usually limited to a few members that may not adequately sample the internal variability. These type **c** studies generally conclude that there is little to no evidence of Eurasian cooling under global warming, and also little evidence that sea ice loss triggers it (Sun et al., 2016; De and Wu, 2019; Boland et al., 2017). A comprehensive listing of individual studies can be found in, for example, Gao et al. (2015).

Drawing robust conclusions based on model studies is an exercise fraught with challenges and pitfalls. Looking at the same

model output, different researchers could arrive at different interpretations based on their confidence in either the models themselves or fundamental understanding of the physical phenomenon being studied (Shepherd, 2021). Furthermore, we know of some physical processes, especially at high latitudes, that our best models do not properly simulate, as discussed in Section 3.3. It is often unappreciated that, regarding Eurasian cooling, we are currently in the unfortunate situation where we have incomplete knowledge about both the climate processes under investigation (e.g. sea ice loss as a trigger for Eurasian cooling)

and the models' abilities to simulate these processes.

### 3.2   Key concepts from selected model studies

This section presents selected modelling studies to open a more in-depth discussion about the methods employed, the results obtained, and the interpretations and conclusions that have been drawn. In doing so, we will elucidate the current disagreements and hint at some opportunities to reconcile apparently conflicting results from previous studies. Because the type **b** approach

(perturbation experiments with uncoupled AGCMs or coupled models) has given rise to most of the results and debate on this topic, we focus on these studies.

There is a long history of the type **b** approach being used to study atmospheric and surface climate responses to sea ice variability and change (e.g., Newson, 1973). Uncoupled perturbation experiments use AGCMs forced with "low" sea ice conditions vs "high" or normal conditions, and sometimes also the corresponding warm vs cold ocean conditions. The forcing

fields may be based on different periods (Peings and Magnusdottir, 2014; Nakamura et al., 2015) or individual years with low or high sea ice extent (Kim et al., 2014; Mori et al., 2014), or on the transient evolution of SST and sea ice concentration (SIC) or thickness over the historical period (Mori et al., 2019; Ogawa et al., 2018; Liang et al., 2020). Many authors report that models can simulate some amount of Eurasian cooling, or more precisely the Warm Arctic Cold Eurasia (WACE) pattern (sometimes called the Warm Arctic Cold Siberia or WACS pattern). However, most if not all of the experiments produce weak





signals compared to the observed trend, especially averaged over runs from many different models (multi-model mean), or over

large numbers of runs (often called "members") from the same model (a "large ensemble" mean). This could indicate that the
observed trend is largely a result of internal atmospheric variability, which is known to mediate or possibly even overwhelm
ice-driven circulation signals (Peings, 2019; Liang et al., 2020). Note that in experimental setups with constant forcing ("high"
versus "low" or normal ice conditions), this internal variability will be underestimated, thus boosting the relative strength of
the ice-driven signal. Obviously, the observed trend in the WACE pattern itself contains variability that is not statistically
related to sea ice, and whether this is well represented in models continues to fuel discussions (Mori et al., 2019; Screen and
Blackport, 2019). Many studies further suggest that if a portion of WACE variability is related to sea ice at all, it is a temporary
phenomenon under global warming (Overland et al., 2016). For example, no WACE response was found in AGCM experiments
forced with prescribed, end-of-century (2090-2100) sea ice under a high-emission scenario (RCP8.5 from CMIP5), although a
strong negative Arctic Oscillation was simulated, producing cold conditions in northwestern Europe (Peings and Magnusdottir,
2014).

Coupled perturbation studies are a subset of the type **b** approach that use various techniques to induce sea ice loss rather
than simply prescribing it. These techniques include sea ice albedo reduction, nudging, and flux adjustment (see explanation
of techniques and their limitations in Screen et al., 2018, Box 1). An important message is that these coupled perturbation
setups capture the full, global impacts of Arctic sea-ice loss because unlike uncoupled perturbation setups, they allow ocean
adjustments to the induced sea ice changes along with subsequent feedbacks to the atmosphere. In their synthesis of six
such studies, Screen et al. (2018) report some consistent but very weak atmospheric circulation responses among the model
experiments (Deser et al., 2015; Blackport and Kushner, 2016, 2017; McCusker et al., 2017; Oudar et al., 2017; Smith et al.,
2017), including a strengthening of the Aleutian Low and Siberian High, a weakening of the Icelandic Low, and an equatorward
shift of the midlatitude westerly wind belts. The stronger Siberian High shifted westerlies would contribute to Eurasian cooling,
but the authors comment that such regional signals are expected to be even weaker in coupled perturbation experiments than
uncoupled because of cancellation between the thermodynamical and dynamical effects of sea ice loss (see also e.g., Deser
et al., 2016; Chripko et al., 2021, and section 4).

Trends may be addressed by using transient historical sea ice forcing in modelling experiments, and these tend not to pro-
duce notable Eurasian cooling signals (Ogawa et al., 2018; Mori et al., 2019). For example, McCusker et al. (2016) produced
600-year long control and perturbation simulations that compared climatological surface conditions over the Arctic from be-
fore and during the Eurasian cooling period. Probability density functions (PDFs) of Eurasia temperatures show no change in
the probability of colder temperatures between the two experiments, and a similar PDF for their large ensemble of coupled
GCM experiments show that the observed cooling sits in the tail of the distribution, suggesting it is a randomly occurring
rare event. Similar results are reported by Ogawa et al. (2018) using 5 AGCMs, each running 20-member transient ensem-
bles, and following the approach of Screen et al. (2013) for observed vs climatological SST forcing (GREENICE Ensemble,
https://greenice.b.uib.no/). Again the ensemble mean shows no trend for Eurasian cooling and the observations sit in the tail
of the distribution (their Figure 2), though it should be noted these results are for the period of 1982-2014 which shows very
weak Eurasian cooling (c.f. Figure 2).





One interesting feature is the vertical extent of the warming associated with sea ice loss, which appears to be linked to the strength of the Eurasian cooling signal. In the Ogawa et al. (2018) study, some individual members show trends in the WACE pattern comparable to observed trends, and all of these members have deep tropospheric heating over the Arctic compared to the rest of the GREENICE ensemble. He et al. (2020) further analysed the GREENICE ensemble (type **b**) as well as a large number of historical simulations (type **c**; specifically, the CMIP5 multi-model ensemble and a single-model large ensemble

CESM-LE), and confirmed that deep tropospheric polar warming is indeed an important feature separating simulations that have strong Eurasian cooling from those that do not. The analysis focuses on interannual variability rather than trends and shows that deep warming is closely connected to the occurrence of Ural Blocking. He et al. (2020) take this to indicate that the deep warming is internally generated, because all analyzed members have identical forcings, whether surface (GREENICE) or radiative (CMIP5 and CESM-LE). Their results are consistent with other type **c** studies that find instances of cold Eurasian

winters or trends in CMIP-class models, and conclude that Eurasian cooling most likely stems from large-scale circulation variability internal to the models (Kelleher and Screen, 2018; Peings, 2019; Blackport et al., 2019). The importance of deep Arctic warming for the WACE pattern and the inability of sea ice alone to create this deep warming is thus a common thread (Deser et al., 2016; Labe et al., 2020).

    The magnitude and even the presence of Eurasian cooling relies upon a precise balance between thermodynamic and dynamic

processes, as discussed further in Section 4. The relevant questions for the real world then are exactly what this balance is and what role sea ice has played in altering it over the last few decades.

### 3.3   Limitations of models

There are a few key limitations of climate models that may explain why a model study could fail to reproduce observed trends. It can be because the model is missing, or insufficiently resolves, one or more of the processes which cause the observed trend.

For example, in wintertime we typically have a shallow, stably stratified atmospheric boundary layer over land, such as over the continental interior of central Eurasia. It has been demonstrated that climate models do not resolve this layer, that they tend to overestimate the amount of mixing in the atmospheric boundary layer, and so underestimate trends in the surface air temperature (Davy and Esau, 2014). Another example is the proposed stratospheric bridge mechanism, whereby sea ice reduction triggers wave activity to propagate into the stratosphere, weakening the polar vortex and changing stratospheric circulation. This in

turn feeds back to the low troposphere, inducing a negative phase of the NAO and increased frequency of Ural blocking, which results in Eurasian cooling (Kim et al., 2014; Nakamura et al., 2016; Zhang et al., 2018a). The ability of the models to accurately simulate the stratospheric bridge may therefore be an important ingredient, as suggested by studies showing stronger Eurasian cooling signals in models that accurately reproduce the evolution of the stratospheric vortex (Garfinkel et al., 2017), and weaker or non-existent Eurasian cooling signals when tropospheric-stratospheric interactions are suppressed (Zhang et al.,

2018a; Nakamura et al., 2016). In fact, this is true of any process that may, if incorrectly represented, act to conceal links between the Arctic SIC and Eurasian cooling by altering the North Atlantic-Eurasian atmospheric circulation (Peings and Magnusdottir, 2015, 2016; Siew et al., 2020) or to reduce the predictable component of internal variability (Scaife and Smith, 2018; Liang et al., 2020; Strommen and Juricke, 2021).





A second potential issue is that an underlying model bias may prevent the model from being able to simulate the observed
trend. For example, if an observed trend is related to the location of sea ice retreat and not simply the magnitude, then a
model bias in the sea ice extent within the region related to the observed response could prevent the model from simulating
the observed trends. CMIP5-era models were generally biased towards over-estimating sea ice extent and didn't capture the
rapid retreat of sea ice in the Barents-Kara sea region, which has been suggested as playing a key role in Eurasian cooling
(Stroeve et al., 2012; Petoukhov and Semenov, 2010; Outten and Esau, 2012). Furthermore, every model's biases will be
slightly different. Thus, even if models can capture a certain teleconnection mechanism between sea ice loss and Eurasian
cooling, the details of this teleconnection (pattern, timing, etc.) would be expected to vary from model to model, making it
difficult to interpret results from a multi-model ensemble-mean (García-Serrano et al., 2017).

Aside from limitations of the models themselves, the design of the model experiment may also introduce limitations. For
example, any model simulation is a discrete sample of an underlying PDF of the potential model outcomes. If we have too
few samples of the underlying PDF (too short a simulation or too few ensemble members) we may not see any instances of
strong signals or trends, despite the fact that the model is capable of reproducing them. As discussed previously, many model
studies utilise atmosphere-only GCMs, which lack interactions between atmosphere, ocean, sea ice, and land that are provide
important low-frequency variability (Bretherton and Battisti, 2000; Barsugli and Battisti, 1998) and feedbacks on the response
of the large-scale atmosphere (Screen et al., 2018; Deser et al., 2015, 2016). Finally, there are studies that employ some form
of surface flux adjustment in coupled models to alter sea ice conditions. While these models are more physically realistic
than atmosphere-only models, they also tend to have larger climatological biases and the experiments do not conserve energy
(Screen et al., 2018).

Despite the shortcomings of climate models, the wholesale dismissal of modelling results is not warranted. Climate mod-
els are continuously improving in their ability to simulate Arctic climate and show increasingly significant skill in making
predictions over Eurasia (previously identified as an area of low reliability, see, e.g., Weisheimer and Palmer (2014)) and the
Arctic in winter on seasonal timescales (Weisheimer et al., 2020; Davy and Outten, 2020). Interestingly, the prediction models
with the smallest biases are not necessarily the best model in terms of predictive skill. Models also appear to reproduce the
observed co-variability between sea ice and Eurasian winter temperatures during the Eurasian cooling period (Outten et al.,
2013; García-Serrano et al., 2017; Cheung et al., 2018; Blackport et al., 2019), suggesting that many of the physical processes
linking the regions are well represented.

## 4   Sea-ice driven or internal variability: Reconciling previous studies

### 4.1   Common ground

In looking at climate signals associated with sea ice loss, some features emerge consistently across most observational (section
2) and modelling (section 3) studies. The most robust are the hemispheric warming and moistening, concentrated in the Arctic,
which are primarily due to thermodynamic changes. As the atmosphere is exposed to open ocean where sea ice retreats, more
solar radiation is absorbed at the ocean surface, stored in the ocean mixed layer during the ice-free warm season, and released





to the Arctic atmosphere during the cold season (Screen and Simmonds, 2010b, a; Screen et al., 2012, 2013; Suo et al., 2016; He et al., 2018). Arctic cloudiness and precipitation also increase, associated with the local warming and moistening (Suo et al., 2016; Kopec et al., 2016; Chernokulsky et al., 2017; Bintanja, 2018).

Other signals associated with sea ice loss vary in detail from observational to modelling studies, or across different modelling studies. Some of these signals are also thermodynamic in origin, for example, the thickening of the atmospheric boundary layer in the Arctic, which seems to be reproduced in model experiments as long as there is appreciable SST warming along with sea ice reduction (Screen et al., 2012; Ogawa et al., 2018; Blackport et al., 2019). The thickening is associated with local dynamical changes such as decreases in sea level pressure (SLP) and geopotential height over the Arctic polar cap, which are
somewhat sensitive to how much sea ice loss occurs in a given experiment. Further from the Arctic, identifying circulation responses to sea ice loss is more challenging (section 3.2). The clearest signals come from coupled experiments with very large ice perturbations and substantial ocean warming, equivalent to projected changes near the end of this century. These generally agree on a strengthening of the Aleutian Low, a weakening of the Icelandic low, and an equatorward intensification of the eddy-driven jet streams (c.f. Screen et al., 2018, and references therein). In uncoupled experiments perturbed with observed sea ice
anomalies or trends, the responses are far less consistent (Mori et al., 2014; McCusker et al., 2016; Honda et al., 2009). For the Eurasian cooling-related signals documented in section 2 (the cold anomaly over the continent itself and the anticyclonic anomaly over the Urals), their sign and location are within the model spread in studies using model simulations that adequately sample the internally generated variability of the climate system, i.e. large ensembles of perturbation runs. However, the signals are very weak, not only compared with observations but also compared with the spread due to (observed and simulated) internal
climate variability (McCusker et al., 2016; Screen, 2014).

## 4.2   Diverging interpretations

What caused the strong Eurasian wintertime cooling in the observational record? The wide array of studies reviewed here have applied a variety of statistical and diagnostic analyses to reanalysis data sets, perturbation experiments, and climate model simulations in an attempt to answer this question. Many have concluded that sea ice is the primary cause of observed Eurasian
cooling; nearly as many have concluded that internal variability is the primary cause. How do we make sense of this?

There is little doubt that internal variability includes a teleconnected pattern with warmer Arctic temperatures, reduced Barents sea ice, and colder Eurasian temperatures compared to normal conditions, in both the real world and climate models. Evidence comes from studies examining interannual variability in detrended observations, detrended historical simulations, and long control simulations, as well as trends in large ensembles of transient historical simulations. Physical arguments from
lead-lag relationships, surface flux signatures, and comparisons between coupled and uncoupled simulations all indicate an important role for internal variability in generating synchronous anomalies over the Barents Sea and Eurasia (Sorokina et al., 2016; Blackport et al., 2019; Blackport and Screen, 2021). Further support comes from the recent disappearance of Eurasian cooling (Figure 2) along with its associated midlatitude circulation signals (Blackport and Screen, 2020), despite continued sea ice loss in the Barents Sea and continued Arctic warming. The conclusion, therefore, is that internal variability must have
contributed to the observed cooling.





The more complicated issue is whether sea ice loss has made (or will make) cold Eurasian winters more likely, and here disagreements arise. The crux is that Eurasian cooling in modelling experiments of various types is non-existent or very weak compared to the observed 15-year trend starting in the late 1990s: it seems to emerge as a statistically significant signal only under certain setups and given a large enough sample size (Mori et al., 2014; Chen et al., 2016; McCusker et al., 2016). This

weak signal-to-noise applies to all the modelling studies we are aware of, which means that the disagreement over the role of sea ice actually lies in the interpretations of the studies rather than the reported results themselves.

Studies concluding that sea ice loss is an important driver of observed Eurasian cooling argue that the models and/or experimental setups are imperfect and hence unable to reproduce the observed signal. Conversely, studies concluding that sea ice loss has had minimal influence on Eurasian winters argue that the large internal variability of the atmosphere will sometimes result

in periods of strong Eurasian cooling and sea ice decline, in which context the 1998-2012 Eurasian cooling is a foreseeable but uncommon occurrence in both model simulations and the real world (true for models, unknown for the real world because our observational record is not long enough). It is unfortunately impossible to point to one clear conclusion. The studies that provide the greatest physical insight into this issue look for indicators of ice-forced or atmosphere-forced Eurasian cooling and assess how models represent the partitioning of these signals compared to observations. However, this has proven challenging

given the complexity of the coupled climate system (c.f. Cohen et al., 2020, and references therein).

For example, Mori et al. (2019) examined the covariability in Eurasian winter temperatures (represented by the WACE pattern of variability) between reanalysis and uncoupled simulations forced with observed sea ice and ocean conditions. They found through maximum covariance analysis that the observed and simulated WACE patterns co-vary, concluding that WACE variability and Eurasian cooling are ice-driven but too weak in models. However, the analysis approach was found to alias

internal variability into the co-varying mode. Redoing the analysis to fully isolate the distinct covariance patterns suggests that Eurasian cooling is in fact atmosphere-driven (Zappa et al., 2021), in agreement with the surface-flux based studies mentioned above (Screen and Blackport, 2019). An alternative interpretation of the covariance analysis is that ice-driven WACE variability dominates in the real world and the models completely fail to produce it. But if so, Zappa et al. (2021) point out that it would be entirely inappropriate to use models to argue for the causal role of ice as well.

While model errors may contribute to differences between observed and simulated Eurasian cooling, it is unfair to discount modelling results as simply wrong. The Eurasian cooling trend has not continued into the recent decade, forcing the community to reevaluate what exactly the "observed" signal is. Furthermore, the bulk of the evidence indicates that the physical mechanisms linking sea ice and Eurasian winter temperatures are in fact present and operating in climate models (Blackport et al., 2019; Screen and Blackport, 2019; Blackport and Screen, 2021; Siew et al., 2021).

### 440  4.3  Looking through the same lens

A key to reconciling the "ice-driven" versus "internal variability" viewpoints of Eurasian cooling is to recognize two distinct components in the linkage between sea ice and Eurasian winter temperatures.





1. The first is a dynamic component that plays a part in both the ice-driven and internal variability mechanisms. An anti-cyclonic circulation anomaly positioned near the Urals advects cold air from the Arctic towards the interior of Eurasia, and warmer air from the midlatitudes over the Barents Sea. Such a circulation anomaly can arise naturally as part of atmospheric internal variability, in which case it promotes both Eurasian cooling and Barents sea ice loss. It can also arise in response to Barents sea ice loss, in which case it promotes Eurasian cooling and amplifies the ice loss itself.

2. The second component is thermodynamic and is associated with the ice-driven response. Sea ice loss induces warming, both directly by exposing more open ocean to the atmosphere, and indirectly through associated ocean and water vapour feedbacks that lead to warming of SSTs outside of the Arctic, in what has been called a "mini" global warming effect (Deser et al., 2015; Blackport and Kushner, 2017).

Thus, we have a dynamic component that cools Eurasia and a thermodynamic component that warms Eurasia. Under strong climate change, local warming in Eurasia can dominate over the dynamically driven cooling effect, and/or the advected Arctic air itself may become much warmer. Such is the case in coupled perturbation experiments, which include the "mini" global warming mentioned above (Screen et al., 2018), or in uncoupled experiments where both SST and sea ice are changed (Sun et al., 2016; Ogawa et al., 2018). The sea ice perturbation may still induce an anticyclonic circulation anomaly that acts to cool Eurasia, but this is partially or completely offset by thermodynamically induced warming (Petrie et al., 2015; Semmler et al., 2016; Blackport and Kushner, 2017; McCusker et al., 2017; Smith et al., 2017; Chripko et al., 2021). Additionally, the overall background warming itself (especially at lower latitudes) contributes to deeper warming of the troposphere (Sellevold et al., 2016; Blackport and Kushner, 2018; He et al., 2020) that allows for strong midlatitude responses in circulation and snowfall (Dai and Song, 2020). The atmospheric response may also depend on the specifics of the ocean feedbacks in ways that are not yet fully understood (Deser et al., 2016). But more central to the debate is that there are two ways the climate system can produce the WACE pattern: through internal variability, where the associated WACE signal is large and robust in both models and observations, or through sea ice loss, where the associated signal is less systematic and more difficult to discern. Whether Eurasian cooling occurs ultimately depends on a balance between dynamic and thermodynamic processes, and how strong the ice-forced dynamical response is in any given situation relative to internal variability.

## 5 Proposed framework

The Cohen et al. (2020) paper presented two divergent views, that either sea ice or internal variability was the primary cause of the observed wintertime Eurasian cooling, each supported by evidence from many previous studies. The evidence in many cases relies on determining whether the inferred or simulated teleconnection is statistically significant compared to some envelope of background variability. But given the relatively short observational record and the large envelope, one may question how meaningful it is to view the issue in terms of statistical significance. Shepherd (2021) suggests that the divide in the published literature might not be as stark if we adopted a "plural, conditional" perspective that accounts for the relatively poor state of prior knowledge, i.e., the uncertainty in the hypothesis that Eurasian cooling is a causal effect of ice retreat (Wallace et al., 2014; Shepherd, 2016).



In light of these uncertainties we can re-examine the observed Eurasian cooling with an aim to reconciling internal variability with sea-ice forcing. On the internal variability side, a sequence of naturally occurring variations in the climate system could align to have given rise to the observed cooling trend over Eurasia without any teleconnection from retreating Arctic sea ice. Longer term observations (Figure 5a) as well as large model ensembles suggest that an unforced 15-year cooling trend

comparable to the observed Eurasian cooling (-2.2 K/decade from 1998/99-2012/13) is certainly possible, but should be a rare event (Figure 5b, note location of dark blue line with respect to the grey lines at the base of the histogram). The rarity of the event need not remain constant through time though. Due to the complex and coupled nature of the climate system, any number of factors could alter the envelope of internal variability, and hence the probability of such a cooling trend. Thus, a teleconnection between sea ice and mid-latitude temperatures may be partly responsible for enabling an otherwise rare

cooling event to have occurred. Viewed from another angle, the internal variability alone is capable of producing a wide range of possible wintertime climates, including a 15-year period of wintertime cooling as seen from 1998 to 2012. But such a strong cooling trend could be more likely with declining sea ice, via either the dynamical effect or an overall enhancement in temperature variability over the mid-latitude continent.

With only one realisation of Earth's climate, we may never get a proper handle on the "true" sampling distribution of climate

variability. Studies suggest that it is overestimated in models for some features like regional temperature trends (McKinnon and Deser, 2018; Deser et al., 2020), underestimated for other features like low-frequency fluctuations in the extra-tropical circulation (Bracegirdle et al., 2018; O'Reilly et al., 2021), but generally comparable for Arctic-midlatitude teleconnections (Kolstad and Screen, 2019; Siew et al., 2021). In an instrumentally derived time series of Eurasian wintertime temperatures (area-averaged temperature over 40-60 N and 60-120 E for DJF seasons) covering the last 70 years, it is clear that the 1998-2012

Eurasian cooling signal is rather special (Figure 5). Exactly how special though depends on the timeframe of interest.

In the context of the full 70-year record (Figure 5a), the Eurasian cooling signal sits at the very edge of the observed distribution (Figure 5b). Using a simple autoregressive (AR1) model that captures the main features of the observed distribution (Figure 5c, orange bars), however, we see that the probability of the observed Eurasian cooling event may change substantially over time. The AR1 model is defined by

$$T_{i+1} = \rho_{-1} T_i + \sigma \sqrt{1 - \rho_{-1}^2} N(0,1), \tag{1}$$

where $\sigma$ is the standard deviation of Eurasian DJF temperature, $\rho$ is the partial lag-1 autocorrelation, and N(0,1) is Gaussian noise (see details in the Supplemental Material). Given the standard deviation of the entire 70-year record, the model produces a distribution of 15-year temperature trends that is comparable to reanalysis (compare orange bars in Figure 5e with Figure 5b), but with longer tails, as expected given the much larger sample size. In addition, results from the CESM Large

Ensemble are shown in grey bars. Using the same 70-year period as the reanalysis and 40 available ensemble members, the distribution is nearly identical to that from the AR1 model, providing confidence that the AR1 model captures important aspects of temperature variability in the region of interest.

The important parameter is the standard deviation $\sigma$, which ranges from 1.171 K to 2.323 K on decadal time scales (Supplemental Figure S4), representing a factor of two difference in the expected spread from internal variability. The observed





Eurasian cooling is more likely during periods with relatively large standard deviation (e.g. Figure 5c, $\sigma$=2.145 K, event is above the 1.46th percentile) than periods with smaller standard deviation (e.g. Figure 5d, $\sigma$=1.171 K, event is at the 0.01st percentile). Similar approaches, applied to phenomena like the North Atlantic Oscillation, have been used to demonstrate that increased chances of extreme trends are linked to larger variability of the trend process (Eade et al., 2021).

Any forcing that alters the background variability of wintertime Eurasian temperatures also alters the probability of observing
a Eurasian cooling-type event. Sea ice retreat has been suggested to increase variability (Li et al., 2015), and hence, make such an event more likely. However, this effect should be assessed as one of many possible factors (e.g., ENSO, PDO, AMO, stratospheric variability, aerosols) that can influence Eurasian winters using appropriate methodologies for quantifying the causal contributions (Kretschmer et al., 2021; Shepherd, 2021). Even if any direct causal effect of sea ice on Eurasian winters is weak or intermittent, it may still be important for its ability to shift the spread due to internal variability and explain intermodel
differences in future projections of climate change (Overland et al., 2021; Kretschmer et al., 2018, 2020; Siew et al., 2020). Rather than asking whether the strong 1998-2012 cooling trend in Eurasia was due to the observed sea ice loss, a more constructive way forward is then to ask whether the cooling trend was more likely given the observed sea ice loss. This framing sets up distinct roles for internal variability and sea ice loss. The former is the roll of the dice that allows for strong cooling trends, strong warming trends and everything in between. The latter may load the dice in favour of cooling trends, an effect
that is expected to wane as global warming intensifies.

## 6 Conclusions

The objective of this synthesis has been to elucidate the disagreement in studies and proposed explanations related to the wintertime Eurasian cooling of the last decades. Our approach has been to lay out the key features in the observations and from the vast array of modelling studies, and to reconcile, where possible, the similarities and differences at the heart of the ongoing
scientific debate.

As highlighted by Cohen et al. (2020), there is an apparent divide in the literature. Many observational studies support a role for forcing from sea ice reduction and Arctic amplification in giving rise to the observed Eurasian cooling, while many modelling studies suggest that Eurasian cooling results from internal variability alone. However, this generalization is not so clear-cut. There are observation-based studies, especially those examining the direction and timing of surface fluxes, supporting
the narrative of internal variability, and there are modelling studies, especially those based on simplified representations of the climate system under idealized perturbations, providing evidence for Eurasian cooling response to sea ice loss. In our framework, these two proposed viewpoints are not mutually exclusive. Any mechanism for Eurasian cooling involving forcing from the Arctic must act concurrently with internal variability, which is always present in the climate system. When considered from a probabilistic standpoint, the internal variability provides an expected spread in temperature trends over Eurasia, with
the observed 1998-2012 trend standing out as a possible but extremely unlikely event. Any forcing that acts to shift or broaden this distribution towards negative values would thus increase the odds of strong cooling trends. If such a forcing relies on a mechanism that is poorly represented in climate models (for example, a stratospheric pathway that is not well reproduced in




low-top climate models), the likelihood of seeing strong cooling events in models would be different than in the real world. Furthermore, sea ice is just one of many factors (ENSO, PDO, AMO, stratospheric variability, aerosols, etc.) that can influence

Eurasian winters. Its ability to "load the dice" in favour of wintertime cooling may have important regional impacts, but should also be viewed in the context of large internal variability and ongoing global warming.

While the recent Eurasian cooling was a transitory event which appears to have now passed (Figure 2), studying Arctic to mid-latitude teleconnections offers avenues to improve regional climate predictions and to expand our fundamental under-standing of climate variability. With the need to bring models and observations together for many applications in the future,

the observed Eurasian cooling documented here provides an interesting case study. It is our intent in this synthesis to highlight the agreements as much as the differences in existing studies, providing a balanced summary of the current state of knowledge for those entering the field. This also allows us to rephrase the scientific debate, asking whether the observed cooling was more likely given the observed sea ice loss.

Frameworks have been proposed to assess Eurasian cooling in ways that better account for the uncertain nature of both the

teleconnection signal and the underlying physical mechanisms. These include coordinated experiments such as the ongoing Polar Amplification Model Intercomparison Project (PAMIP; Smith et al., 2019) and the use of probabilistic storylines for circulation trends and changes (Shepherd, 2019). In this work, we have presented a probabilistic viewpoint in which internal variability and sea ice retreat are two factors (among many) that affect the chances of strong, regional cooling trends under ongoing global warming.

*Code and data availability.* Code and data for the AR model analysis shown in this paper is available at https://github.com/martin-king/outtenetal2022eurasiacoolingtrends

*Author contributions.* Stephen Outten and Camille Li: Coordination, writing, analysis, and figure creation. Martin P. King, Peter Y.F. Siew, and Lingling Suo: Writing, analysis, and figure creation. Shengping He, Erica Madonna, Richard Davy, Hoffman Cheung, Etienne Dunn-Sigouin, Stefan Sobolowski: Writing. Tore Furevik, Thomas Spengler, and Tim Woollings: Internal review, revisions.

*Competing interests.* Camille Li is a member of the editorial board for Weather and Climate Dynamics

*Acknowledgements.* This work was funded by the Norwegian Ministry of Education and Research through the Bjerknes Centre for Climate Research, with contributions from the Research Council of Norway (DynAMiTe grant 255027, Nansen Legacy grant 276730, visiting grant 310391, BASIC grant 325440).





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



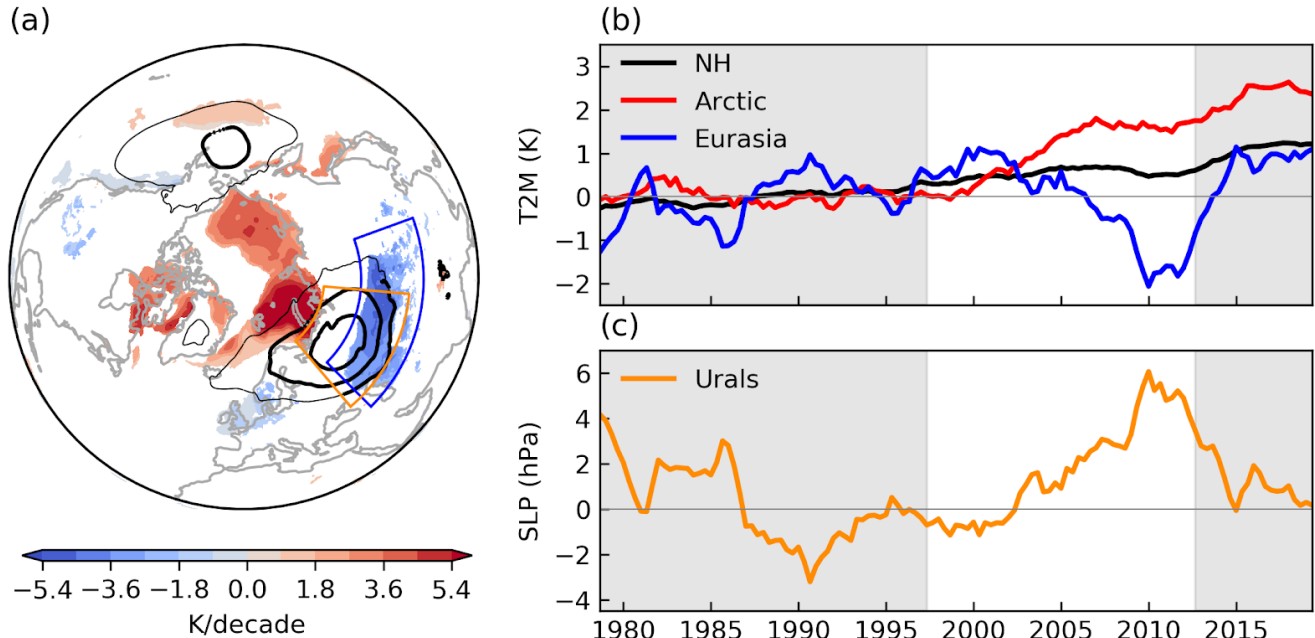

**Figure 1.** The "Warm Arctic-Cold Continent" pattern and the associated surface circulation. (a) Trends in wintertime 2-metre air temperature (shading) and sea level pressure (contours with levels: 3, 6, 9 hPa/decade) in ERA5 reanalysis for 1998 to 2012 over the Northern Hemisphere. Trends for 2-metre air temperature are only shown for locations that are significantly different at the 95% level from the mean Northern Hemisphere trend for the given period. Trends for sea level pressure are shown in thick (thin) contours for locations that are significant (insignificant) at the 95% level from the mean Northern Hemisphere trend for the given period. The blue (orange) box denotes the Eurasian (Urals) region for calculating the area-averaged indices in panel b (c). (b) Indices of area-averaged 2-metre air temperature over the Northern Hemisphere (20-90°N, black), Arctic (60-90°N, red) and Eurasia (40-60°N, 45-110°E, blue) from 1979 to 2019. (c) Indices of area-averaged sea level pressure over the Urals (45-70°N, 40-85°E) from 1979 to 2019. (b-c): The indices are constructed from monthly values from December to February. The monthly climatology has been removed relative to a 1979-1998 period and a five-year running mean applied. The unshaded region corresponds to the period from 1998 to 2012.



**Figure 2.** Trends in DJF-mean 2-metre air temperature in ERA5 reanalysis for periods ranging from 15 to 30 years in length (columns) and for starting years ranging from 1979 to 2005 (rows). The domain shown is 0-150°E and 40-80°N. Trends are only shown when they differ from the Northern Hemisphere trend over the specified period (95% significance level). The inset (bottom right) shows an expanded view of the panel corresponding to the 15-year period highlighted in Figure 1a, from DJF 1998 to DJF 2012.

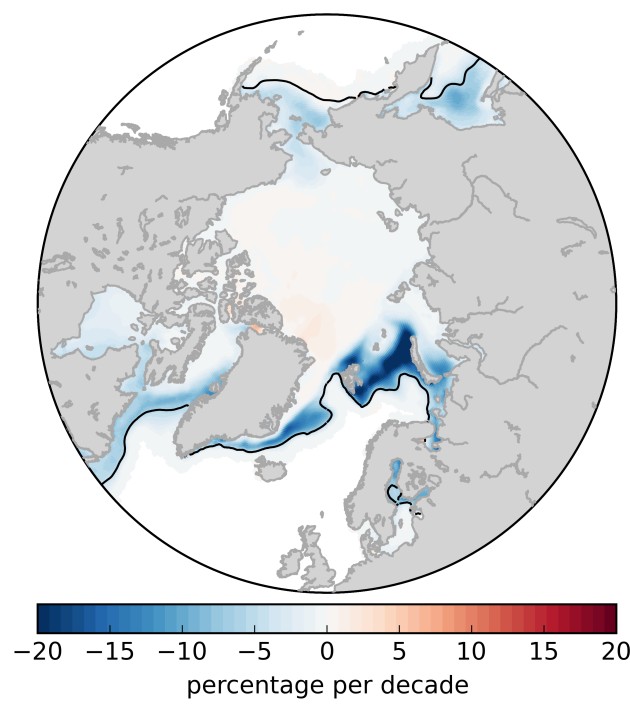

**Figure 3.** The winter (DJF) Arctic sea ice concentration trend in ERA5 over the periods of 1980-2019. Black contour indicates the sea ice extent climatology for 1980-2000.

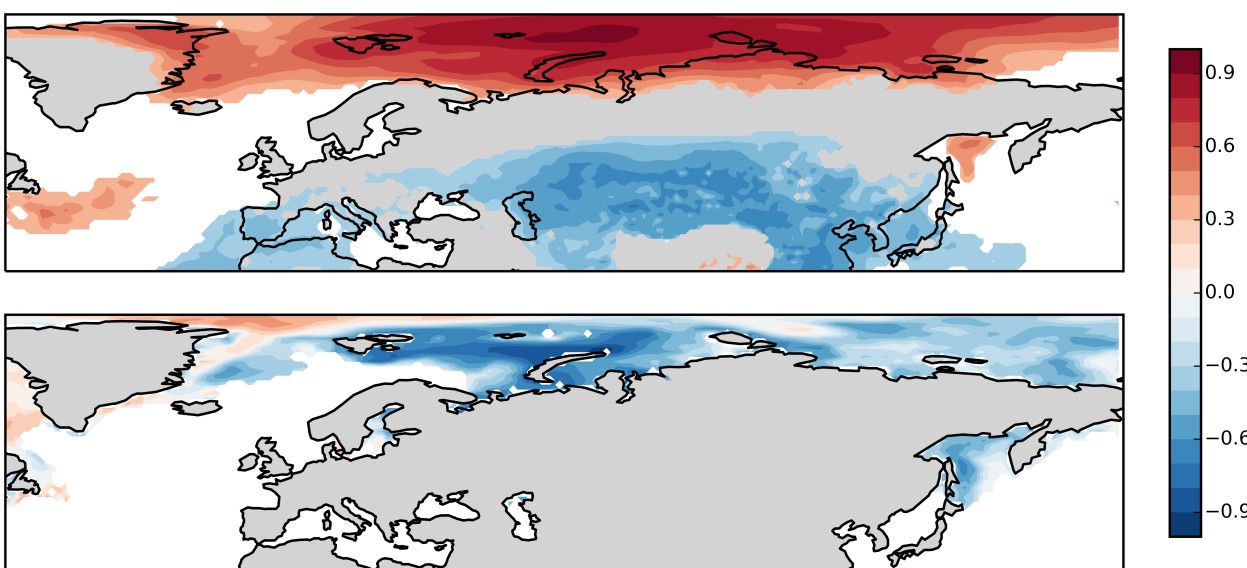

**Figure 4.** Spatial distribution of the first SVD mode calculated over the Northern Hemisphere, mapped as homogenous correlation with surface air temperature (top) and sea-ice concentration (bottom) from ERA5, for the period of 1998 to 2012. Covariance explained is 59% and correlation coefficient is r=0.84. Contour interval is 0.1.

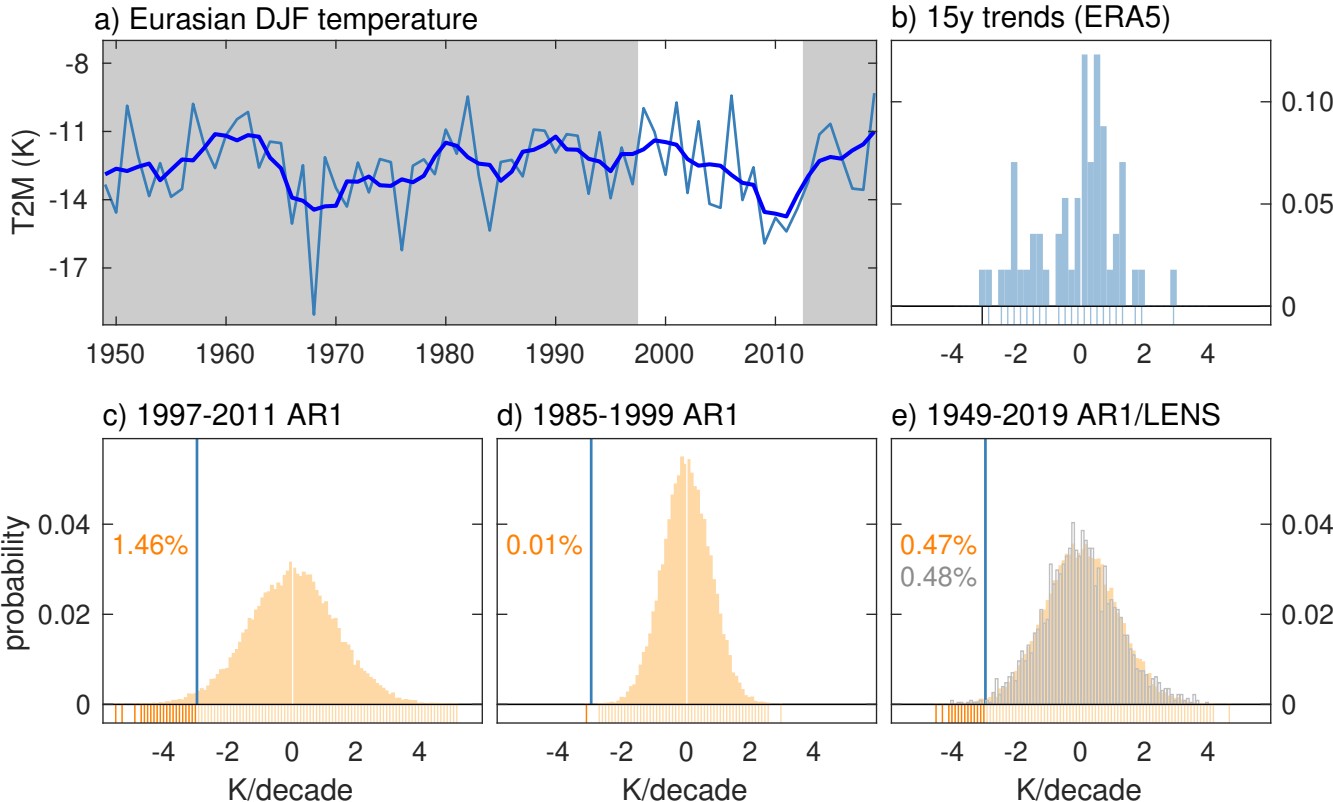

**Figure 5.** Eurasian winter temperature trends. (a) Time series of 2-metre winter temperatures from the ERA5 reanalysis for DJF 1949/50 to DJF 2019/20, averaged over 40-60°N and 45-110°E (box shown in Fig. 1). The thin line shows winter averages, and the thick line has a 5-year running mean applied. The period with the strongest 15-year cooling trend (pattern of temperature anomalies shown in the Fig. 2 inset) is 1998-2012, where the year corresponds to the December of each DJF winter season. (b) Histogram of observed 15-year temperature trends. (c) Probability distribution of 15-year temperature trends from an AR1 model with relatively high standard deviation ($\sigma$=2.145 K taken from the period 1997-2011). The vertical blue line indicates the 1998-2012 trend (-2.21 K/decade), the percentage indicates where the 1998-2012 trend falls in the distribution, and the bars at the bottom are an alternative visualization of the spread of the distribution, where darker lines show cooling trends stronger than 1998-2012. (d) Same as (c) but with low standard deviation ($\sigma$=1.171 K taken from the period 1985-1999). (e) Same as (c) but with $\sigma$=1.817 K for the entire period 1950-2019. Grey bars show the probability distribution of 15-year temperature trends from the CESM Large Ensemble (LENS), using the simulation period 1950-2019 and all 40 available members.