# Peer review of "Reconciling conflicting evidence for the cause of the observed early 21st century Eurasian Cooling"

_Weather and Climate Dynamics, 2022_

## Author Response (AR1)

**FINAL AUTHOR COMMENTS**

wcd-2022-32 Reconciling conflicting evidence for the cause of the observed early 21st century Eurasian cooling (Outten et al.)

We thank Judah Cohen and two anonymous reviewers for their time in reviewing the manuscript and their thoughtful input. Below are their comments in black with our responses in blue. Line numbers given for individual changes are referring to the tracked-changes version of the paper. Line numbers are not given for all changes as some responses required small alterations and clarifications in multiple places throughout the paper.

**Response to Reviewer 1 - Judah Cohen**

The manuscript tries to reconcile numerous observational analysis studies and model sensitivity experiments of Arctic mid-latitude linkages that offer a wide range of conclusions on whether recent observed Eurasian cooling is related to and/or in response to sea ice loss or the two are coincident in time but unrelated physically and the cooling can be attributed to internal variability of the atmosphere. The authors argue that the modeling and observational studies are not at odds or that we must necessarily conclude either/or that Arctic sea ice loss either contributes to Eurasian cooling or that the cooling is related to internal variability only. Instead, the authors argue that different conclusions can all be at least partially correct and that the cooling can be related to multiple factors at once.

I thought that the discussion of the subject and uncertainty was comprehensive and offered a novel way or at least crystallized the idea better than previous published papers on the subject to frame the debate that can help advance the discussion and how to reconcile all the disparate conclusions. Though it might be obvious that sea ice forcing and natural variability can operate simultaneously, or is not a new idea (this was the thesis of Overland et al. 2021), I think the authors expounded on this idea better than previous studies that I am familiar with. I also thought that the manuscript advanced the conversation beyond Cohen et al. (2020) where it was argued that the conclusion whether sea ice melt can force continental cooling can be grouped into observational and modeling studies and instead the authors here argue that the groups are in reality much more mixed. I also thought that the discussion around Figure 5 that even if sea ice only changed the standard deviation of surface temperatures, that alone can increase the probability of Eurasian cooling even if it the sea ice doesn't directly force an atmosphere response conducive to Eurasian cooling.

One suggestion is to maybe shorten the text. I thought that it was a long windup for the punchline. I thought that the concluding remarks were well stated and valuable and I think that it would benefit the reader to get to these important conclusions sooner. But I will not go as so far to suggest text to remove and I leave it totally up to the authors.

We appreciate the idea to get the punchline faster, especially for a reader who is very familiar with the field. However this is a review paper and we would like to ensure it contains all of the details so as to be useful for people who know little about the subject. We agree that some of the text is protracted and we have streamlined the text where possible throughout. We have also added suggestions in the intro paragraph of section 2, as well as signposts throughout sections 2 and 3, that readers familiar with the subject matter may skip certain parts (e.g., sections 2.1, 2.2, 3.1, 3.3) without loss of continuity, thus shortening their reading by four to five pages.

**Line 88 and elsewhere in sections 2 and 3.**

As is reflected in some of my minor comments below, I do take issue with this idea that with the inclusion of the most recent observational data, the empirical analysis has come into agreement with the modeling studies that there is no large-scale atmospheric circulation forced response to sea ice variability and that Eurasian cooling has all but disappeared. I am not raising my own paper to require that the authors cite it but rather because it is the paper that most readily comes to my mind that shows trends and the observed relationship between sea ice and large-scale circulation variability over the full reanalysis period. As seen in Cohen et a. 2021 Figure 3, the relationship between sea ice and atmospheric variability remains robust (at least in scale and based on statistical significance) and from Figure S6 Eurasian winter cooling is seen over 41 years of reanalysis and that Eurasia is a clear outlier to the widespread warming elsewhere across the Northern Hemisphere.

We note that the general impression of reviewer 1 that we have gone too far in saying that there is "no" large-scale forced response to sea ice variability is somewhat at odds with the general impression of reviewer 3, who feels we have gone too far in saying that there is some forced response. In any case, we did not mean to convey the message that including the most recent observational data leads us to the conclusion of "no" large-scale forced response, although we see that this misunderstanding was likely due to some of less than careful wording on our part (e.g., L221-222 in the original manuscript where we should not have specified we are talking about trends, and not interannual variability as shown in the correlations of Fig. 3 in Cohen et al. 2021). Hopefully, in the revisions outlined below, we have been able to clarify some of the statements that each of these reviewers object to by providing more explanation and a more balanced discussion on the studies that have backed the various viewpoints. For example, in the response to comment #1 below, we clarify why we say that the Eurasian cooling \*trend\* (specifically, over what time scale) has largely disappeared.

Thanks for pointing out Cohen et al. 2021, which is certainly relevant and which we missed. It is a useful reference for extreme winters and the role of the stratosphere.

**Small refinements throughout the paper.**

I have some more minor comments below and I recommend that the manuscript be accepted pending minor revisions.

Minor comments:

 Line 116 – I agree that the Eurasian cooling trend peaked around 2012/13 and has since dampened but I think to characterize it as passed is an overstatement. To expect a perpetual cooling trend is unrealistic given the rapid rise in global temperatures. Winter temperatures in the region of interest remain cooler relative to other regions of Northern Hemisphere and overall cooler than model forecasts. Do you know what else peaked in 2012? September Arctic Sea ice melt, would the authors claim the era of Arctic sea ice melt has "passed?"

There is no doubt that a long-term Eurasian cooling trend shows up over the reanalysis period - Figure S6 of Cohen et al. 2021 shows it for the 41-year period 1980-2021, and the rightmost column of Figure 2 in our manuscript shows it for 30-year periods from 1981 to 2020. But from the columns further to the left, it is clear that the long-term trend is dominated by strong, shorter-term trends that are concentrated around the mid-1990s to

mid-2010s. Our statement pertains to these strong cooling trends, and this has been clarified. As the reviewer points out, cooling trends should weaken and eventually disappear under global warming. We have attempted to account for this, because significance is calculated compared to the Northern Hemisphere-averaged warming trend - therefore, even weak cooling trends will show up in Figure 2. This detail is mentioned in the caption but other reviewers missed the information as well, so we have made it more explicit in the text.

Regarding sea ice, we did not mean to suggest that the era of Arctic sea ice melt has passed. The difference between sea ice (top) and Eurasian temperature (third from bottom), in terms of their variability versus trend, is nicely illustrated in the attached figure of Blackport & Screen 2020 (already in the reference list). We have made sure this is clear in the text, and have referenced this figure more explicitly where appropriate.

**Changes on lines 121-124, and in Observational Summary, lines 238-241.**

**2. Line 253 – Not sure why only the reference to GAO (2015) is listed, can the authors include a more up to date reference?**

The Gao et al. paper does contain a very nice table over several pages which lists a large number of the key studies along with the models and forcings used, but it is indeed an older summary. We have rephrased the sentence to make clear that the Gao et al. paper provide a good list of studies up to 2015, but that there are many newer studies, and we provide a short list of newer references.

Refined the explanation of the contents of the Gao et al. paper on lines 286-287, and added references to multiple other studies during the rewrite of the preceding paragraph on lines 259-281.

3. Lines 407-408 – again I feel that this statement and conclusion presented as fact – "the recent disappearance of Eurasian cooling along with its associated midlatitude circulation signals" is misleading.

This was due to a combination of our wording (not careful enough), some misunderstandings related to Fig. 2, and not indicating where support for parts of this statement come from - all of which are addressed in other parts of this response. We have revised the statement now. Please see responses to earlier comments, plus the response to reviewer #2 comment L219-220.

**Revised lines 457-458.**

4. Lines 435-436 – I think to say "it is unfair to discount modeling results as simply wrong" is overly strong. I think a better way of saying something similar like "it is unfair to attribute differences between observed and simulated Eurasian cooling to model errors or deficiencies only." The exact wording is not important, but I don't think anyone would argue that the models are deficient to be useless.

We like and agree with the wording you have kindly suggested and have replaced the statement in the paper accordingly.

**Changed lines 489-490.**

5. Lines 437-438 – it is my opinion that the physical mechanism can exist in the models and yet the models can still miss much if not all the Eurasian cooling forced by the iceatmosphere mechanism especially when looking at the ensemble mean.

This is a good point and we agree. The point is raised in section 3.2, but discussion surrounding it has been expanded based on comments from reviewer #3. We have now also added a comment earlier in the subsection 4.2 that the choice of forcing, experimental setup and ensemble size are very important factors in whether a given experiment will "get" Eurasian Cooling.

Revised second paragraph of section 3.2 on lines 301-328, and added lines 466-468.

**Judah Cohen**

References:

Smith, D.M., Eade, R., Andrews, M.B. *et al.* Robust but weak winter atmospheric circulation response to future Arctic sea ice loss. *Nat Commun* **13**, 727 (2022). https://doi.org/10.1038/s41467-022-28283-y

Cohen, J., L. Agel, M. Barlow, C. I. Garfinkel, I. White. 2022: Arctic change reduces risk of cold extremes—Response, *Science*, **375** (6582), 729-730, DOI: 10.1126/science.abn8954.

**Response to Reviewer 2**

This paper analyses the extensive scientific debate around the role of Arctic warming (more specifically, localized warming from Arctic sea-ice loss) in recent Eurasian wintertime cooling trends. Although there have been many overview papers on this topic published in recent years, this one is particularly good. That may be because the authors are people who I would not place in one camp or the other. The paper is comprehensive, balanced, and reflective. As well as providing a very nice and useful synthesis of recent studies, it offers a reframing of the question that should provide a constructive way forward on what everybody agrees is an important area of scientific research. The paper correctly notes that a problem with much of the current debate is (i) the failure to acknowledge that a definitive yes-no answer is not possible given all the uncertainties involved, and (ii) the assumed dichotomy between the mean forced response and internal variability, as if they were separable. Yet the wintertime Arctic is arguably the place where the internal variability is most likely to change in response to climate change, and this separation is least defensible.

Effectively, the authors are suggesting a hypothesis that observed trends over a particular period may be primarily attributable to internal variability, but that the internal variability may have changed because of Arctic warming in such a way that the probability of such trends has increased. That is a very novel way of framing the question at hand. It will almost inevitably involve different hypotheses (or storylines) for how the internal variability might have changed, which can be compared in terms of their consistency with data. For this purpose, the proposed emphasis on the distinction between the thermodynamic and dynamical aspects of the problem will be very useful, as the different hypotheses will almost certainly be on the dynamical side. That distinction is not new in climate-change science, but I believe is new (or at least under-utilized) in this particular context.

Overall, this is an excellent and timely paper. I feel that for far too long the debate in this area has been largely sterile, encouraged by certain journals which seem to like papers with titles that are unconditional, to 'stir the pot'. This paper can help set a new tone, and lead to better science. I am happy to recommend acceptance largely as is, with just minor revisions.

**Minor comments**

line 6: I'm not sure that "coincidental" is the right word. From my understanding, the word can have either an inferential interpretation (by chance) or a descriptive one (at the same time). The latter has no causal implication either way, so is presumably not what is meant here. Do you really mean by chance, or rather that the two features are correlated because of a common driver (atmospheric variability)? (Correlation may not imply causation, but unless it really is by chance, it has to reflect causation somewhere in the system.)

This is correct, coincidental is a bad choice of word here. Upon revision, we have removed the last part of the sentence containing the word since it was unnecessary. Thank you for pointing this out.

**Change on line 6.**

Caption to Figure 1, line 1: Wouldn't it be better to refer to WACE rather than WACC here, since it is Eurasia that is singled out?

This makes good sense and we have changed the text to read "Warm Arctic-Cold Eurasia".

**Changed text in caption for Figure 1.**

Caption to Figure 1, lines 4-5: Should be "significantly different", not "significant". And not sure what is meant by "insignificant"; do you really mean that there are no trends (of at least 3 hPa/decade) anywhere else on the map?

You are correct and it has been changed to say "significantly different". Note that the figure shows trends which are significantly different from the mean NH trend, which is positive. This was written in the caption but has also been added to the text now We've double-checked the SLP trend contours and they are correct. The 15-year period over which the trends are calculated smooths things out quite a lot in most other regions. Figure 5d from Mori et al. 2019 for the period 1995-2014 DJF using ERA-Interim, pasted below, shows a similar result (contour lines -2, -1, 1, 2, 3, 4, 5... hPa/decade).

**Changed text in caption of Figure 1.**

**Figure 2: I suspect that some reviewers might complain about the size of these postage-stamp images, but for me it works!**

We did consider and test other ways of plotting this figure, e.g. plotting every other start year and period length, thus reducing the number of plots by a factor of four. However, after numerous attempts we decided this was the best way to plot the figure to provide a good overview of the changes in trends, which is the intention of the plot. Since many readers these days will read the paper electronically, the figure can be zoomed in to see some of the details in the individual plots if that is what they are interested in. So far, everyone has liked the postage-stamp plot, perhaps because people are already familiar with the small multiples concept from Ed Hawkins' global warming maps.

**No changes.**

lines 175-179: I think it is only fair to refer to Kretschmer et al. (2021 BAMS) [in your reference list] here, who so far as I know were the first to point out the two rather different definitions of teleconnection in the AMS Glossary.

Thank you for pointing this out. We agree that reference should be made and have therefore added a statement and the appropriate reference.

**Added text and reference in lines 196-197.**

lines 219-220: The statement "the circulation trends and Eurasian cooling itself have not continued into the most recent decade, while sea ice loss and Arctic warming unequivocally have" seems overcooked. For Arctic warming, it is contradicted by your earlier statement on line 124 that "the Arctic warming trends disappear in the period starting in 2005", which is clearly apparent in Figure 2. For sea ice, it is contradicted by your Figure S3 (as well as by other such figures which one can find on the NSIDC web site). You need to tone this paragraph down.

We did not explain the "disappearing" Arctic warming properly - it is in fact Arctic amplification that disappears. Thanks to the reviewer for noticing this. We have clarified this throughout the paragraph with the contradictory statement (originally L124) and also in other places discussing the Fig. 2. See response to reviewer #1 comment #1 for more detail. We have added a version of Fig. 2 without the significance masking to the supplementary material and included it below for your convenience.

Regarding sea ice, while there was a peak in sea ice loss around 2012 (as with the Eurasian Cooling trend), it has indeed continued - in a much more consistent manner than EC. In fact, the statement in L219-220 is well supported by Blackport & Screen 2020's figure pasted above (see reviewer #1 comment #1), which we realize we should have referenced here as well. However, we have modified the wording - together with the clarifications described above, this will hopefully give readers a more accurate picture.

Revised paragraph on lines 127-140, and elsewhere when referring to Figure 2. Included a new Figure as Supplemental Figure S2, as shown below. Revised text in Observational Summary, lines 237-241.

---

## Author Response (AR2)

**RESPONSE TO SECOND REVIEW**

wcd-2022-32 Reconciling conflicting evidence for the cause of the observed early 21st century Eurasian cooling (Outten et al.)

We'd like to thank Reviewer 3 for their time and comments once again. Below are their comments in black with our responses in blue.

**Response to Reviewer 3 - Second Review**

The authors have made improvements to many aspects of the manuscript. Overall, the review and the interpretation of the literature in sections 1-4 is excellent and will be very important given the conflicting conclusions across individual studies. However, I still think the authors may be over-interpreting the results discussed in section 5.

The authors are assuming that the changes in standard deviation seen over 15 year periods in the observed time series (and the odds of a cooling trend) are real, inherent changes (whether forced by sea ice or something else). They never explicitly state this, but it is heavily implied by the analysis and with statements like "with some periods being more susceptible to strong cooling than others" (L12-13 in abstract). However, the standard deviation will vary over short periods across the time series simply by chance without any changes in the real, inherent variability. For example, if you created synthetic data with an AR1 model with a fixed standard deviation, there will be (by chance) 15 year periods with a higher and lower standard deviations because of the small sample size. This, of course, does not mean that the trends were inherently more or less likely during these periods. Similarly, you can't claim that any differences in the observed standard deviation of real changes in the inherent variability, without additional evidence.

Related to the above issues, I may not have been clear in my original comment about the possibility of the trend causing a larger standard deviation. I was not referring to the AR1 model where the standard deviation is fixed. I was referring to the observed time series where the causality can go in both directions. Even if there is no change in the inherent variability, a trend (whether caused by random chance or as part of a forced response) will cause a larger standard deviation.

The authors need to support their claims that the changes in variability are real with additional evidence, or they need to tone down these statements and add caveats/discussion about these issues.

Thanks to the reviewer for these additional comments, and we accept the points raised by the reviewer. From the beginning, the analysis in section 5 was only ever meant to demonstrate some concepts that could be useful in thinking about Eurasian cooling and should not to be viewed as evidence of some mechanism. We felt we had communicated this better in the last revision, but upon re-reading the section, we see how the reviewer could be concerned about possible misunderstandings. In light of this, we have made additional edits throughout the section and to the abstract, and hope that they clarify our intent.

As the reviewer suspects, there is substantial uncertainty in the standard deviations estimated from 15-year periods. The difference between the "extreme" values from the reanalysis data is significant just below the 90% confidence level (the 90% confidence interval for the minimum value of 1.171 K is [0.9, 1.7] K and for the maximum value of 2.145 K is [1.65, 3,13] K, so they overlap slightly). The same conclusion can be drawn from the AR1 model, though the bounds are slightly different. We need to be cautious about deriving statistical properties from 15-year periods, echoing the point regarding temperature trends made early on in the manuscript. These points are now included more explicitly in section 5.